# Endoplasmic Reticulum Stress in Bronchopulmonary Dysplasia: Contributor or Consequence?

**DOI:** 10.3390/cells13211774

**Published:** 2024-10-26

**Authors:** Tzong-Jin Wu, Michelle Teng, Xigang Jing, Kirkwood A. Pritchard, Billy W. Day, Stephen Naylor, Ru-Jeng Teng

**Affiliations:** 1Department of Pediatrics, Medical College of Wisconsin, Suite C410, Children Corporate Center, 999N 92nd Street, Milwaukee, WI 53226, USA; twu@mcw.edu (T.-J.W.); mteng@mcw.edu (M.T.); xgjing@mcw.edu (X.J.); 2Children’s Research Institute, Medical College of Wisconsin, 8701 W Watertown Plank Rd., Wauwatosa, WI 53226, USA; kpritch@mcw.edu; 3Department of Surgery, Medical College of Wisconsin, 8701 Watertown Plank Rd., Milwaukee, WI 53226, USA; 4ReNeuroGen LLC, 2160 San Fernando Dr., Elm Grove, WI 53122, USA; billy.day@rngen.com (B.W.D.); snaylor@rngen.com (S.N.)

**Keywords:** bronchopulmonary dysplasia, proteostasis, endoplasmic reticulum stress, unfolded protein response, chaperone, cellular senescence, *N*-acetyl-lysyltyrosylcysteine amide

## Abstract

Bronchopulmonary dysplasia (BPD) is the most common complication of prematurity. Oxidative stress (OS) and inflammation are the major contributors to BPD. Despite aggressive treatments, BPD prevalence remains unchanged, which underscores the urgent need to explore more potential therapies. The endoplasmic reticulum (ER) plays crucial roles in surfactant and protein synthesis, assisting mitochondrial function, and maintaining metabolic homeostasis. Under OS, disturbed metabolism and protein folding transform the ER structure to refold proteins and help degrade non-essential proteins to resume cell homeostasis. When OS becomes excessive, the endogenous chaperone will leave the three ER stress sensors to allow subsequent changes, including cell death and senescence, impairing the growth potential of organs. The contributing role of ER stress in BPD is confirmed by reproducing the BPD phenotype in rat pups by ER stress inducers. Although chemical chaperones attenuate BPD, ER stress is still associated with cellular senescence. *N*-acetyl-lysyltyrosylcysteine amide (KYC) is a myeloperoxidase inhibitor that attenuates ER stress and senescence as a systems pharmacology agent. In this review, we describe the role of ER stress in BPD and discuss the therapeutic potentials of chemical chaperones and KYC, highlighting their promising role in future therapeutic interventions.

## 1. Introduction

Human complexity and variability have limited our understanding of the pathobiology of disease causality, onset, and progression [1]. It is salutary to consider that each of us consists of approximately ~37 trillion cells, and approximately 50% are microbial constituents of the “human microbiome” [2]. We consist of ~200 different cell types [3], possess ~20,000–25,000 genes [4], 100,000–2,000,000 proteins (including isoforms) [5], and >3000 metabolites [1]. Even an organ such as a lung is biologically complex. There are ~40–60 cell types, including epithelial, endothelial, fibroblast smooth muscle, nerve, and assorted immune cells [6]. The adult human lung contains approximately 480 million alveoli to effect efficient oxygen and carbon dioxide gas exchange. They provide a large surface area of ~70 square meters (the size of a tennis court) for efficient gas exchange to meet the body’s oxygen demands [7]. In contrast, newborns have only ~20–50 million alveoli in their lungs at birth [8]. This limited understanding due to complexity has resulted in the lack of safe and efficacious therapeutic drugs for treating numerous specific disease conditions [1]. One example of such a complex disease state with meager therapeutic options is bronchopulmonary dysplasia (BPD) [9].

BPD is the most common pulmonary complication in premature infants [10]. Roughly 30–50% of extremely premature neonates develop BPD [11]. More than 15,000 new cases of BPD are diagnosed each year in the United States, with annual costs estimated to be more than USD 2.5 billion [12]. The impact of BPD on parents and their families, medical services, and society is significant [13]. Prematurity with surfactant deficiency and a limited number of alveoli, a condition called respiratory distress syndrome (RDS), is the initial causative event for BPD. With the increased trend of premature births and improved medical management, more and more extremely premature neonates (<29 weeks of gestation) are now surviving, which contributes to the unchanged annual BPD prevalence. Implementing clinically tested BPD treatment, such as intramuscular vitamin A injection [14] and early postnatal caffeine treatment [15], does not reduce the BPD rate, exemplifying the poorly understood pathobiological BPD processes.

BPD’s three major postnatal contributors are supplemental oxygen, mechanical ventilation, and infection. All these contributors cause excessive oxidative stress (OS). The poorly developed antioxidant capacity in premature neonates makes them susceptible to OS. Judicious oxygen use, early surfactant therapy, gentle ventilation strategy, aggressive nutritional support, early postnatal caffeine treatment, and vitamin A injection have been widely accepted as standard management for premature neonates. The unchanged or slightly increased prevalence of BPD indicates, however, room for improvement, and new therapeutic modalities should be explored. A common approach for developing new therapeutic interventions is identifying novel pathobiological processes in the targeted disease condition. For example, the role of endoplasmic reticulum (ER) stress in BPD has recently been reported by us [16], and this review, in part, assesses the potential of targeting pathway/network constraints in ER stress to treat BPD effectively.

The ER is a subcellular organelle synthesizing surfactants [17], proteins, lipids, and cholesterols [18]. It intimately interacts with other organelles, essential for cell homeostasis. The ER is sensitive to OS, however, which can result in ER stress [19]. ER stress has been implicated in several lung disorders, including hyperoxia (HOX)-induced lung injury [20]. Our recent report was the first to show increased ER stress in human BPD lungs [16]. We further identified that ER stress is interrelated with autophagy, apoptosis, sterile inflammation, and cellular senescence [21]. All these associated pathologic changes can inhibit alveolar formation and reduce the lung growth potential. In animal studies, we then demonstrated that caffeine and chemical chaperones attenuated the severity of HOX-induced rat BPD by ameliorating ER stress. Later, we identified the crucial role of myeloperoxidase (MPO)-induced OS in BPD [22]. We demonstrated decreased ER stress in BPD rat lungs by treating rat pups with the reversible MPO inhibitor N-acetyl-lysyltyrosylcysteine amide (KYC) with improved alveolar formation [21]. This new observation raises the possibility of using MPO inhibitors in ER stress-mediated disorders.

Despite these findings, it is essential to recognize that ER stress is not typically the primary cause of disease onset in BPD or other major disease states. Rather, ER stress represents a significant downstream consequence of various initiating factors, such as OS and inflammation [9]. This distinction between continuum and causality has considerable ramifications for therapeutic development. Specifically, it raises important questions about whether targeting ER stress directly is the most effective strategy or if addressing the upstream causes of ER stress might yield better therapeutic outcomes. This nuance will be explored in depth throughout this review as we examine the role of ER stress in BPD within the broader context of disease pathogenesis. In this review, we focus specifically on BPD. We will start by describing the structure and function of the ER, followed by how OS elicits ER stress and the role of ER stress in BPD. We will then discuss the therapeutic potential of chemical chaperones and MPO inhibitors in BPD treatment.

## 2. Endoplasmic Reticulum (ER)

The ER is a unique subcellular organelle with a continuous membrane structure that accounts for more than half of the cellular endomembrane system [23]. It forms a complex channel system from the nucleus to the extracellular space. The ER intimately interacts with other subcellular organelles through specialized contact sites. The ER provides a rapid transit system with multiple hubs for material storage within the cell. There are two interchangeable shapes of the ER: tubule and sheet. Under stress, such as nutritional deprivation, tubular ER will transform into sheet-like structures. The loss of ER structural integrity is profound [24] and leads to the following:(i)Disruption of protein folding, resulting in ER stress and the unfolded protein response (UPR);(ii)Altered lipid metabolism;(iii)Perturbance of calcium homeostasis;(iv)ER–Golgi trafficking inefficiencies;(v)Mitochondrial dysfunction, leading to pronounced oxidative stress (OS);(vi)Cellular stress and associated co-morbidities.

### 2.1. Structure of ER

There are two types of ER: rough ER (RER) and smooth ER (SER). Each has different physical and functional characteristics. RER is seen throughout the cytoplasm but is preferentially clustered around the nucleus and the Golgi apparatus. Ribosomes attached to the RER are called membrane-bound and are responsible for protein synthesis. The RER’s membrane is continuous with the nuclear envelope’s outer membrane [25]. SER is a network of tubules and sacs, and its function is devoted mainly to metabolism and lipid synthesis. Other activities of the SER include calcium storage, detoxification, and synthesizing carbohydrates and cholesterols [26]. A critical function of the SER is to support mitochondrial activity and regulate ER–Golgi traffic [27]. Different types of cells have different ratios of the two types of ER depending on their activities [28].

#### Regulation of ER Shape

The ER is a dynamic subcellular organelle with a complex structure maintained by various integral membrane proteins and interactions with other organelles and the cytoskeleton [29]. Its tubular structure contains a high curvature area, requiring specialized proteins to maintain it.

### 2.2. ER Structural Proteins

#### 2.2.1. Reticulons and REEPs

The reticulon (RTN) family is the most studied one that regulates the tubular ER’s morphology with highly curved edges. These proteins contribute to the bending of the membrane by forming a transmembrane hairpin topology that acts as a wedge [28]. RTN works with other proteins to shape the ER structure. The most extensively studied RTN is RTN4, also called Nogo. There are three RTN4 isoforms (A, B, and C). Only RTN4A and RTN4B work to maintain the ER structure [30]. RTN4A and its receptors are predominantly expressed in the central nervous system [31]. RTN4B, on the other hand, is widely distributed and interacts with FAM134C to promote ER membrane curvature [32]. RTN4A/B depletion results in the loss of ER tubulation [33]. Other abundant ER-resident proteins that help shape the ER structure include the receptor expression-enhancing proteins (REEPs) family that localizes explicitly to tubules and the edges of sheets [34].

#### 2.2.2. Reticulon Receptor

Some RTNs have corresponding receptors. The most extensively studied RTN receptors are related to RTN4A, such as ErbB3 [35], S1PR2 [36], and DP1 [37,38]. These receptors have specific downstream activities unrelated to structural maintenance except DP1, which is known to assist the tubular shape of the ER. RTN4B has a receptor usually described as the Nogo-B receptor (NgBR) or nuclear undecaprenyl pyrophosphate synthase 1 (NUS1) in the literature. NgBR is required for RAS activation [39] and dolichol synthesis [40]. As dolichol synthesis is a critical first step for protein N-glycosylation, its depletion can impair angiogenesis [41]. Interestingly, NgBR knockdown in endothelial cells affects the morphology of ER and mitochondria (personal communication). The mechanism by which NgBR affects the morphology of the ER requires further exploration, but the evidence of ER stress developed after NgBR knockdown in pulmonary artery smooth muscle cells provides us with a plausible explanation [42].

##### Atlastins

At least three isoforms of atlastins, or atlastin GTPases, have been reported (ATL1, ATL2, and ATL3). Atlastin-1 (ATL1) is primarily produced in the central nervous system, especially the spinal cord. ATL1 sustains the branched tubular ER network [43] by catalyzing homotypic membrane fusion [44]. ATL2 is involved in the organization of the Golgi apparatus and protein homo-oligomerization. ATL3 is required to form the network of interconnected tubules of the ER adequately. The ATL family also remodels the ER for ER-phagy [45].

### 2.3. Functions of the ER

The ER is part of the transportation system of eukaryotic cells, which allows some substances to be transported rapidly without requiring energy expenditure. The ER also provides a place for substance storage or synthesis.

#### 2.3.1. Protein Synthesis, Folding, and Modifications

The RER has ribosomes on its surface that synthesize proteins with a signal sequence (leader sequence or localizing signals) that directs nascent peptides or proteins to the ER for processing. The short signal sequence is present at the N-terminus (or occasionally non-classically at the C-terminus or internally) of most newly synthesized proteins destined for the secretory pathway. The KDEL or HDCL signal sequences are C-terminus tetrapeptides that prevent proteins from leaving the ER [46]. Once nascent proteins enter the ER, BiP/GRP78 chaperone protein binds them to help the folding and assembly through a primary oxidase, Ero1, which creates the oxidizing environment and catalyzes disulfide bond formation [47]. The adequately folded proteins will then be modified by UDP-glucose/glycoprotein glucosyl transferase into a glycoprotein before secreting outside the cell or going to other organelles, like the Golgi apparatus or lysosomes, for further modification.

#### 2.3.2. Lipid and Cholesterol Synthesis

The ER membrane produces lipids for itself and other organelles, including mitochondria and peroxisomes. The SER also synthesizes cholesterol, a precursor to steroid hormones [18]. Surfactant is a lipid–protein mixture that reduces the surface tension to open the alveoli. Surfactant is produced in the ER and is processed through the Golgi apparatus of the type 2 alveolar cells (AT2) [17]. After synthesis, the surfactant is packaged within the lysosome-derived lamellar body before releasing into the alveolar sac. Disturbed ER function may thus contribute to surfactant dysfunction [20].

#### 2.3.3. Calcium Regulation

The ER is the principal site of intracellular calcium storage. Calcium channels, calcium transporters, calcium pumps, and calcium-binding proteins precisely regulate cytosolic calcium. Calcium depletion can induce ER stress and cell death [48]. The ER–mitochondrial contact microdomain, mitochondria-associated membranes (MAM), allows the ER to provide calcium through the voltage-dependent anion channels for supporting mitochondrial activity [49]. Without this support under ER stress, mitochondria cannot function appropriately, and electron-transport chain uncoupling to generate OS will ensue.

#### 2.3.4. Detoxification

The ER, especially the SER, performs drug and poison detoxification in liver cells. The SER contains and synthesizes specialized enzymes that convert lipophilic drugs into hydrophilic derivatives that can be expelled from the body [50]. The most typical enzyme is the cytochrome P450 (CYP450). CYP450 enzymes are membrane-bound hemoproteins that work with CYP450 reductase and cytochrome b5 to metabolize harmful compounds through oxidation [51].

### 2.4. Organelle Communication

The ER is sometimes considered the central organelle in cells. It maintains cellular homeostasis through physical contact, molecular interactions, chemical exchanges, signal transmissions, and inter-organelle regulation with other organelles [52]. The ER forms a specialized membrane contact domain with other organelles to achieve inter-organelle communications [23].

#### 2.4.1. ER and Mitochondria

MAMs are specialized contact sites between the ER and mitochondria. MAMs are rich in proteins and enzymes that regulate various physiological processes, including calcium transfer, lipid synthesis and transport, mitochondrial dynamics, glucose homeostasis, apoptosis, autophagy, and the formation and activation of an inflammasome [53]. More than 1000 proteins have been detected in the MAM, demonstrating its extreme complexity and importance to cell homeostasis [54]. Mitofusin 2 has been identified as the tethering protein between the ER and mitochondria that lynchpins the two organelles into proximity [55]. The MAM allows the exchange of essential metabolites for many homeostatic processes, including organelle quality control, cellular metabolism, and signaling [56]. Mitochondrial function is critical for adequate angiogenesis [57]; thus, disturbed MAM function will impair angiogenesis [58].

Mitochondrial fission is part of a continuous cycle of fission and fusion that helps mitochondria maintain a healthy pool for metabolic demands [59]. This process’s purpose includes mitochondrial quality control, the formation of new mitochondria, being part of the cell division process, and mitochondrial DNA segregation. When necessary, mitochondrial fission facilitates apoptosis under high levels of stress. Mitochondrial fission also separates damaged mitochondria from functional mitochondrial networks. The ER helps shape the mitochondrial network, marking sites for mitochondrial fission [60].

Abman first coined the vascular hypothesis in 2001 to describe how impaired angiogenesis contributes to alveolar simplification in BPD [61]. Other experts in BPD research have widely accepted this concept [62]. We thus can expect a disruption of ER–mitochondria interaction to contribute to BPD.

#### 2.4.2. ER and Nucleus

The ER interacts with the nucleus through physical connections [63], signaling [64,65], and nuclear positioning [66]. The outer membrane of the ER forms a continuum with the outer nuclear membrane, with multiple junctions to supply the nuclear envelope with proteins and lipids. In addition to the physical connection, the downstream signalings of ER stress can mediate nuclear responses (described below). The ER helps the nucleus establish the asymmetric nucleo-cytoskeleton connection and allows proteins to be transported to the nucleus without spending energy [29].

#### 2.4.3. ER and Other Organelles

The ER interacts with other organelles (Golgi apparatus, lysosomes, endosomes, peroxisomes, vacuoles, secretory vesicles, and plasma membrane) through direct membrane contact sites. These interactions are vital for regulating organelle biogenesis and dynamics, lipid homeostasis, and calcium dynamics. The ER has abundant contact sites with the plasma membrane and interacts with ribosomes at the RER. The extensive interaction between the ER and other organelles is thus vital to cell homeostasis [52,67].

### 2.5. ER-Phagy

Autophagy is an organized, lysosome-mediated survival mechanism of cells during starvation. There are several types of autophagy, including macroautophagy, mitoautophagy, chaperone-mediated autophagy, microautophagy, crinophagy, lipophagy, pexophagy, aggrephagy, lysophagy, and ER-phagy (or reticulophagy) [68]. All types of autophagy will eventually converge in lysosomes to ensure degradation. This process removes unnecessary molecules and recycles the degraded raw materials to maintain homeostasis and cell survival [69]. Several cell death pathways have been reported to interact with autophagy, including necrosis, apoptosis, pyroptosis, ferroptosis, autophagic cell death, and autosis [68].

The ER integrates with other organelles during starvation and changes its morphology into a sheet within 1–2 h as a non-destructive transformation [23]. This rapid morphological change is associated with inhibiting mitochondrial fission, maintaining mitochondrial oxidative phosphorylation, and providing ER homeostasis. Studies in vitro show that when starvation persists for more than 10–16 h, the increased OS and disturbed membrane lipid and fatty acid will result in ER-phagy [70]. Excessive ER-phagy, similar to other autophagy, results in apoptosis and diseases [71] and is also a change observed in acute lung injury [72].

## 3. ER Stress

### 3.1. Proteins Involved in ER Stress

#### 3.1.1. Sensors for ER Stress

There are three specialized proteins, namely, inositol-requiring enzyme 1α (IRE1α), protein kinase R-like endoplasmic reticulum kinase (PERK), and activating transcription factor 6 (ATF6), in the ER that sense overloaded unfolded proteins (Figure 1) [73]. These three sensors are regularly chaperoned by an ER-resident binding immunoglobulin protein (BiP) called glucose-regulated protein 78 (GRP78). BiP/GRP78, or HSPA5, is a family member of heat shock protein 70 (HSP70). BiP/GRP78 binds newly synthesized proteins as they are translocated into the ER and maintains them in a state competent for subsequent folding, oligomerization, and post-translation modification.

When the ER accumulates excessive unfolded proteins (proteostasis perturbation or proteotoxicity), there will not be enough BiP/GRP78 to chaperone PERK, IRE1α, and ATF6. The unchaperoned IRE1α and PERK start to dimerize and become activated through phosphorylation. The unchaperoned ATF6 will be translocated to the Golgi apparatus, cleaved by site-1 and site-2 proteases (SP1 and SP2), and then will be translocated further to the nucleus as a transcription factor. These changes activate downstream signaling to stop non-essential proteins’ synthesis and direct protein synthesis to generate BiP/GRP78. Evidence also suggests that ATF6 is a specialized transcription factor regulating protein quality in the ER [74]. The only purpose for these responses is to resume ER homeostasis. If ER stress persists for too long or fails to resume homeostasis, it will result in autophagy or cell death [75].

#### 3.1.2. Endogenous Chaperone in ER

BiP/GRP78 is the endogenous chaperone and ER stress repressor and chaperones the ER stress sensors. BiP/GRP78 upregulation is one of several cellular coping mechanisms in stressful environments. Besides its role in ER stress, recent studies from Amy S. Lee’s group have expanded our understanding of BiP/GRP78. Under some viral infections, BiP/GRP78 may be translocated to the cell surface to facilitate viral attachment and entry to host cells. It has been shown that the membrane expression of BiP/GRP78 in people with cancer contributes to the increased susceptibility to COVID-19 infection [76]. The surface expression of BiP/GRP78 seems to be associated with tumor initiation and progression [77]. BiP/GRP78 binding with α2-macroglobulin activates at least eight signaling pathways to enhance lipogenesis and adipogenesis that contribute to obesity [78]. BiP/GRP78 can even be translocated to the nucleus to assume the role of a transcriptional regulator [65]. Besides its role in lung cancer, BiP/GRP78 depletion has been shown to suppress alveolar formation [79] and promote lung fibrosis [80].

#### 3.1.3. Endoplasmic Reticulum Oxidase-1 (ERO1) and Protein Disulfide Isomerase (PDI) in ER

Protein folding in ER is a redox process relying on protein PDI and ERO1 (ERO1α or ERO1L, and ERO1B). PDI is considered a specialized endogenous ER chaperone. PDI prevents nascent polypeptides from misfolding and aggression by reducing, oxidizing, and isomerizing disulfide bonds. ERO1 is an oxidoreductase that catalyzes the de novo disulfide bond formation, transfers the disulfide bond to nascent proteins via PDI, and generates hydrogen peroxide (H_2_O_2_). Glutathione peroxidases (GPX7 and GPX8) and peroxiredoxin 4 (PRX4) handle H_2_O_2_ to maintain the optimal ER redox state. Sequential disulfide formations contribute up to 25% of cellular reactive oxygen species (ROS) produced during protein synthesis, which may exceed the mitochondria. Thus, when ERO1 is overactivated or PDI is inhibited, excessive ROS generation can cause ER stress. ERO1 is enriched in the MAM, which may play a role in modulating calcium transfer and MAM dysfunction. Owing to their critical role in protein folding, PDI/ERO1 dysregulation (mainly upregulation) has been demonstrated in cancers, thrombosis, diabetes, and neurodegenerative disorders. Increased ERO1 expression can result in passive calcium flux, which leads to apoptosis [81]. Post-translational modification (PTM) of PDI and ERO1 affects their activities. These PTMs include S-glutathionylation, cysteine nitrosylation, sulfenylation, and phosphorylation. PDI phosphorylation at ser357 results in an open PDI conformation and turns PDI into a holdase that attenuates ER stress [82].

### 3.2. ER Stress Activation

ER stress can be caused by many factors that disrupt protein folding in the ER, including calcium or redox imbalances, hypoglycemia, hypoxia, acidosis, high-fat or high-sugar diet, natural compound (tunicamycin, thapsigargin, or geldanamycin) intoxication, prescribed drugs, nutrient deprivation, viral infections, heat shock, fatty acids, inflammatory cytokines, mutation in proteins entering the secretory pathway, and impaired protein glycosylation or disulfide bond formation [83]. They all lead to the accumulation of unfolded proteins in the lumen of the ER, inducing a coordinated adaptive ER stress response, also called unfolded protein response (UPR).

### 3.3. Survival Mechanisms That Share Similarities with ER Stress

Biologically, ER stress shares similarities with several other mechanisms that also handle proteostasis disorders, such as integrated stress response (ISR), mTORC1 signaling complex, stress-induced granulation, TCF11/Nrf1-NGLY1-DDI2 axis, and pathogen recognition receptors (PRR) [84]. They share similar features, such as (1) dealing with the imbalance between protein influx and degradation (proteostasis perturbation) and (2) resulting in sterile inflammation and autoinflammatory diseases. In this review, we only focus on ER stress. As described below, several reactions occur due to ER stress (Figure 2).

### 3.4. Consequence of ER Stress

#### 3.4.1. Translational Attenuation

PERK activation after its phosphorylation activates the eukaryotic initiation factor 2 (eIF2) signaling. eIF2 activation by phosphorylation inhibits general protein synthesis but promotes the translation of specific mRNA that can support cell survival under stress [85]. This targeted translation attenuation is one of the survival mechanisms that eukaryotic cells adopt.

#### 3.4.2. Inflammatory Response

Evidence from different research has shown that pathways activated under ER stress induce sterile inflammation. All three UPR sensors participate in upregulating inflammatory processes [86]. IRE1α and PERK activations lead to the downstream activation of the NFκB-induced inflammatory response [87]. After cleavage in the Golgi, the cleaved ATF6 activates NFκB through the PI3K–Akt pathway [88]. The NFκB activation explains the neutrophil infiltration in ER stress [89]. Another mechanism ER stress causes is sterile inflammation through the high mobility group box 1 (HMGB1) released from autophagic flux-induced cell death [90]. HMGB1 is a non-histone nuclear protein with potent damage-associated molecular pattern (DAMP) activity. HMGB1 binds to pattern recognition receptors (PRR), such as Toll-like receptor 2 (TLR2), Toll-like receptor 4 (TLR4), and the receptor for advanced glycation end product (RAGE) to activate NFκB-mediated sterile inflammation [91].

#### 3.4.3. Autophagy and Apoptosis

ER stress causes both autophagy and apoptosis. IRE1α phosphorylation activates downstream JNK signaling, while PERK phosphorylation activates ATF4 signaling. JNK and ATF4 activation thus leads to autophagy, apoptosis, and oxidative response [92].

#### 3.4.4. ER-Stress-Associated Degradation (ERAD)

ERAD is a cellular pathway that targets misfolded proteins for ubiquitination and subsequent degradation by the proteasome. Both IRE1α and ATF6 are related to ERAD. IRE1α phosphorylation activates an alternate X-box binding protein 1 (XBP1) mRNA splicing. Spliced XBP1 mRNA leads to increased expression of split XBP1 that causes ERAD. Cleaved ATF6 also activates ERAD. Stressed cells can use ERAD to obtain raw material to regenerate essential proteins to survive [93].

#### 3.4.5. Regulated IRE1α-Dependent Decay (RIDD)

Regulated IRE1α-dependent decay is a process in which phosphorylated IRE1α degrades mRNAs, reducing the amount of non-essential proteins imported into the ER [94]. PERK activation also uses RIDD to degrade mRNA. RIDD will thus help cells to synthesize only essential life-saving proteins.

#### 3.4.6. Increased OS in ER Stress

The active refolding of unfolded proteins generates many reactive oxygen species, since protein folding is highly redox-reaction dependent. The disturbed ER–mitochondrial interaction during ER stress will increase OS due to electron transport chain uncoupling and mitochondrial dysfunction [75].

### 3.5. Disorders That Involve Increased ER Stress—What Can We Learn?

Table 1 presents a list of disorders associated with ER stress, illustrating the commonality of cellular stress being implicated in a wide range of diseases [95]. In Section 2 and Section 3, we briefly discussed the wide range of functions of the ER and the myriad of problems caused by ER stress, respectively. In particular, prolonged ER stress can result in cellular dysfunction and apoptosis [96]. Diseases such as Gaucher, Fabry, and cystic fibrosis involve specific genetic mutations that affect proteins located in the lysosome or plasma membrane, which are processed through the ER. The table highlights that ER stress is not restricted to a single organ or tissue type but is found across diverse conditions, including metabolic disorders like diabetes, neurodegenerative diseases like Parkinson’s, and various lysosomal storage disorders. Importantly, Table 1 underscores the ubiquity of ER stress in cellular dysfunction, demonstrating that regardless of the affected organelle or protein type, ER stress plays a role in the pathophysiology of these diseases [95].

In BPD, the presence of ER stress could provide essential insights into the mechanisms driving disease progression. Since BPD involves both inflammatory and OS responses [97], ER stress may serve as a pivotal link that exacerbates these conditions, mainly through the activation of the UPR. This pathway, if persistently activated, can lead to cell death and tissue damage, further contributing to the chronic lung injury seen in BPD [21]. The recognition that ER stress is a common underlying feature in diverse disease states could lead to the conclusion that therapies aimed at modulating ER stress, which has been explored in other diseases, such as diabetes and neurodegenerative disorders, may hold promise for treating BPD, as well. Targeting ER stress in BPD could help restore protein homeostasis, reduce inflammation, and mitigate oxidative damage, potentially preventing disease progression and improving lung development in affected neonates. However, this presupposes that in BPD, as well as all the other disease conditions listed in Table 1, there is a clear understanding of the complex relationship between ER stress and the disease condition. In the case of BPD, is ER stress manifestation due to causal or consequential effects?

While it is tempting to view ER stress as a promising therapeutic target in BPD, as well as the numerous other disease states of Table 1, the possibility that it is a consequence rather than a cause of the disease raises essential considerations. If ER stress occurs as a downstream event in response to other pathological processes, such as inflammation, OS, or impaired lung development, then targeting it directly may not address the root causes of BPD. In this case, ER stress might represent a cellular response to the broader damaging environment in the preterm lungs rather than a driver of disease onset and progression. This distinction is crucial when considering therapeutic design. If ER stress is merely a byproduct of the ongoing inflammatory or OS damage, interventions aimed at alleviating ER stress might provide symptomatic relief without significantly altering the underlying disease trajectory. In other words, while ER stress-modulating drugs could potentially reduce some aspects of cellular dysfunction, they may not be sufficient to prevent lung injury or improve long-term outcomes in BPD. Thus, focusing on upstream processes, such as inflammation or OS damage, might prove more effective for developing therapies that address the root causes of BPD. Moreover, experimental evidence showing that ER stress occurs after the onset of other pathological events would suggest that interventions should target these earlier events to prevent the cascade of damage that leads to ER stress [16,21,97]. Furthermore, this argument underscores the need for a better understanding of the temporal relationship between ER stress and other pathological processes in BPD. Suppose that ER stress is indeed a consequence rather than a cause. In that case, therapeutic strategies should focus on upstream mechanisms that drive lung injury, with ER stress seen as an essential but secondary factor in the disease’s pathology.

## 4. Treatments That Attenuate ER Stress

In the last section, we raised a question about the potential value of targeting ER stress in general disease conditions, specifically in BPD. Nevertheless, a great deal of effort has been expended on targeting chaperones to alleviate ER stress and the corresponding underlying disease condition. Chemical chaperones, molecular chaperones, and pharmacological chaperones are three terms used in the literature to describe molecules that can ease unfolded protein-induced ER stress. Perlmutter first used a chemical chaperone to describe a small molecule that enhances protein folding and stability [98]. Chemical chaperones can be classified into osmolytes (glycerol, trehalose, and trimethylamine N-oxide), hydrophobic compounds (4-PB), and pharmacological chaperones. Morello et al. first coined the term pharmacological chaperone in 2000 to describe molecules, different from chemical chaperones, which are more specifically designed for specific proteins [99]. Sternberg first described molecular chaperones in 1973 from a bacteriophage study [100]. Heat-shock proteins are the most extensively studied molecular chaperones [101]. GRP78/BiP is the characteristic molecular chaperone within the ER, and it has been considered an ideal target for drug development in treating ER stress [102].

### 4.1. Chemical Chaperones for ER Stress-Related Disorders

Chemical chaperones have been extensively studied for their role in alleviating ER stress by promoting proper protein folding and preventing the accumulation of misfolded proteins. One prominent example is 4-phenylbutyrate (4-PB), an FDA-approved drug initially used to treat urea cycle disorders. 4-PB has since gained attention for its ability to reduce ER stress by stabilizing protein folding and enhancing cellular homeostasis. Studies have shown its efficacy in models of neurodegenerative diseases and metabolic conditions, such as diabetes [103]. Another well-studied chemical chaperone is tauroursodeoxycholic acid (TUDCA). This bile acid derivative has demonstrated protective effects against ER stress in conditions such as amyotrophic lateral sclerosis (ALS) and type 2 diabetes. TUDCA reduces ER stress by inhibiting apoptosis and promoting protein homeostasis [104]. Trimethylamine N-oxide (TMAO), an osmolyte, also acts as a chemical chaperone by stabilizing protein structures and has been studied for its ability to mitigate ER stress in cardiovascular diseases [105]. These examples underscore the putative therapeutic potential of chemical chaperones across a range of disease states characterized by ER stress, highlighting their importance in ongoing drug development efforts.

### 4.2. Chemical Chaperones in Pediatric and BPD Studies

The accumulation of unfolded protein in the ER is the leading cause of ER stress. BiP/GRP78 is the endogenous chaperone that handles unfolded proteins in the ER. ER stress will be ensured when the quantity of unfolded proteins overwhelms the handling capacity of BiP/GRP78. Nature has created some molecules for lower-level creatures that can maintain the conformation of proteins under harsh environments, like UV light, extreme heat, and extreme pH. These molecules stabilize the unfolded proteins non-selectively, facilitate their folding, or prevent protein aggregation in the ER. This group of small molecules, including DMSO, glycerol, deuterated water, betaine, lysine, 4-phenylbutyrate (4-PB), ursodeoxycholate (UDC), and tauroursodeoxycholate (TUDC) [106,107], has been called chemical chaperones. However, not all chemical chaperones have the same performance [108]. As chemical chaperones show protein-stabilizing activity, it is assumed they may help resolve proteostasis perturbation. 

Chemical chaperones have been shown to benefit some disorders. Type 2 diabetes is the most commonly studied disorder with chemical chaperones [109]. Other diseases include inflammatory diseases and protein-folding disorders [95]. Not all compounds that attenuate ER stress can be called chemical chaperones [110]. Chemical chaperones have also been studied to improve disorders in animal models, including Fabry disease, Alzheimer’s disease, and amyotrophic lateral sclerosis [111]. Although 4-PB [112], UDC [113], and TUDC [114] have been used in infants for urea cycle disorder or cholestasis, they have not been studied in ER stress in the pediatric population. Some other chemical chaperones in the literature are sephin1 and salubrinal. Both sephin1 and salubrinal also work on eIF2 to attenuate ER stress [110].

### 4.3. Other Compounds Used in BPD Treatment

Early caffeine treatment and intramuscular vitamin A injection are clinically adopted treatments that decrease BPD. Vitamin D has recently gained considerable attention in BPD prevention and therapy [115]. Some other compounds, including KYC, have been studied in animal models of BPD [9].

#### 4.3.1. Caffeine

Initially, it was considered a respiratory stimulant that might protect premature brains by reducing apnea-induced fluctuation in blood oxygenation. During the secondary analysis, the Caffeine for Apnea of Prematurity (CAP) trial significantly reduced BPD [15]. Although we have shown that caffeine reduces ER stress in HOX BPD rat lungs [116], we did not show how caffeine affects proteostasis. There are also conflicting reports about the effect of caffeine on ER stress.

#### 4.3.2. Vitamin A

Vitamin A and its derivatives (retinoids) have been shown to activate ER stress and, through this mechanism, to increase the effectiveness of cancer treatment [117]. Retinoic acid synergizes with ER stress and OS to kill acute myeloid leukemia cells [118]. One study also showed that activating retinoic acid receptor-related orphan receptor α in adipose tissue will activate ER stress with a pro-inflammatory response [119]. We could not identify any study showing that retinoids can decrease ER stress or act as chemical chaperones. Thus, the evidence suggests that vitamin A decreases BPD, probably not because of its effect on ER stress. 

#### 4.3.3. Vitamin D

Since the randomized controlled trial showing vitamin D prevents BPD in premature infants [120], vitamin D has recently attracted considerable attention in BPD treatment [115]. Its anti-inflammatory and antioxidant properties are considered beneficial for BPD prevention. Studies showed that vitamin D in vitro decreases ER stress induced by thapsigargin, an ER stress inducer, in neuroblastoma cells [121]. Studies showed that vitamin D extends the lifespan of *C. elegans* by reducing ER stress [122]. Since no evidence shows vitamin D stabilizes unfolded proteins, it cannot be called a chemical chaperone, but it seems to decrease the severity of ER stress.

#### 4.3.4. Corticosteroids

The importance of corticosteroids cannot be ignored, as they used to be the standard BPD treatment before we realized that early postnatal systemic corticosteroid treatment is associated with long-term neurodevelopmental deficits [123]. Clinical evidence suggests, however, that systemic corticosteroid treatment after eight days of life might be safe enough to treat premature infants at moderate-to-high risk for BPD [124]. Dexamethasone has been shown to alleviate ER stress in an animal model of inflammatory bowel disease [125]. In human hepatocellular carcinoma cells, however, dexamethasone yielded a contradictory result with enhanced ER stress [126]. We could not find good evidence about how corticosteroids affect ER stress in the lungs, especially regarding BPD.

#### 4.3.5. KYC

KYC is an N- and C-capped tripeptide consisting of lysine, tyrosine, and cysteine [127]. In the presence of chloride and H_2_O_2_, KYC will occupy the iron center of MPO to prevent the formation of HOCl. By occupying the iron center, KYC reversibly converts MPO into a quasi-catalase to remove H_2_O_2_. KYC is converted into thiyl radicals, which then react with thiol-containing proteins in the vicinity during inflammation. This unique chemical reaction results in numerous biological changes to protect neonatal lungs. The protection offered by KYC includes (1) decreasing inflammatory cell infiltration, (2) decreasing ER stress, (3) decreasing apoptosis and cellular senescence, (4) improving endothelial cell function and angiogenesis, and (5) upregulating antioxidant enzymes via stabilizing NRF2 (see our recent review [9]). KYC earns itself the designation as a systems pharmacology agent by its ability to offer multilevel protection [128]. Although we demonstrated that KYC decreases ER stress in BPD rat lungs, we cannot claim its chemical chaperone status without showing evidence that it can stabilize unfolded proteins.

## 5. ER Stress in BPD

ER stress has been demonstrated in several different diseases, such as diabetes, cancer, neurodegenerative disorders, and some lung disorders, including HOX-induced lung injury [20]. The idea that ER stress contributes to BPD stems from the study by Choo-Wing et al., wherein their group showed increased ER stress in HOX mouse pup lungs and alveolar simplification. Although the term integrated stress response was used, the changes can also be explained as ER stress [129]. Lu et al. used premature rat pups (E21) exposed to 85% oxygen to show an increased expression of CHOP and GRP78, apoptosis, and decreased radial alveolar count [130]. Later, our group showed that increased ER stress in HOX-exposed rat pup lungs can be successfully attenuated with early caffeine treatment [116]. In another report, we further demonstrated an increased ER stress in human and rat BPD lungs [16] (Figure 3). These human and animal studies strongly suggest ER stress plays a contributing role in BPD.

The improved alveolar structure with decreased ER stress in the BPD rat lungs by caffeine may not, however, support the role of ER stress in BPD [116]. The concern stems from the fact that caffeine has been reported to induce ER stress [131], although contradictory results have also been reported [132,133]. An ER inducer and a chemical chaperone were added to our animal studies to provide more evidence. Tunicamycin is a mixture of three naturally occurring antibiotics. By inhibiting UDP-*N*-acetylglucosamine-dolichol phosphate *N*-acetylglucosamine-1-phosphate transferase, it blocks the first step of protein N-glycosylation in the ER [134], resulting in ER stress. Rat pups that receive a single intraperitoneal injection of tunicamycin develop a BPD phenotype characterized by alveolar simplification, increased apoptosis, myeloid cell infiltration, decreased capillary count, and increased myeloperoxidase (MPO) expression in the lungs reminiscent of HOX-induce BPD rat lungs [16].

TUDC is an FDA-approved hydrophilic bile-acid derivative used to treat cholestasis with a chemical chaperone activity [135]. It has been used in human neonates for cholestatic liver dysfunction, usually after prolonged total parenteral nutrition [114]. As a complementary study, TUDC was given to HOX-exposed and tunicamycin-treated rat pups. The TUDC treatment successfully reduced ER stress, MPO expression, and myeloid cell infiltration and improved alveolar formation in both HOX-exposed and tunicamycin-treated rat pups (Figure 4). Together with the results from tunicamycin-treated lungs, it was concluded that ER stress does contribute to BPD [16].

Since MPO has been demonstrated to play a crucial role in HOX-exposed rat BPD [16], and hypochlorous acid (HOCl) generated by MPO is a potent reactive oxygen species, we investigated whether inhibiting MPO activity can reduce ER stress in the HOX rat BPD model. Our study used KYC as the MPO inhibitor. The results demonstrate an attenuation of ER stress (Figure 5) [16] and cellular senescence [21] in BPD rat lungs. Interestingly, one of the studies also showed the preservation of AT2 in the BPD rat lungs by both TUDC and KYC [21]. Since the AT2 is considered a residential progenitor cell in the lung after the saccular stage [136], this finding may contribute to a better lung growth potential for the HOX-exposed rat pups. We did not attempt to demonstrate whether KYC has any chemical chaperone activity, but its system pharmacology effects can explain its ability to attenuate ER stress [22].

### 5.1. ER Stress Impairs Angiogenesis

There are at least two mechanisms by which ER stress can impair angiogenesis. First, some growth factors and their corresponding receptors and transporters require proper handling in the ER to obtain full function. One protein crucial to angiogenesis is the vascular-endothelial-growth-factor receptor type 2 (VEGFR_2_) [41]. N-glycosylation of VEGFR_2_ occurs in ER to elicit pro-angiogenic signaling [137]. ER stress will thus impair angiogenesis. Second, mitochondrial activity is supported by intimate support from the ER. Since normal mitochondrial function is required for angiogenesis, ER stress can again disturb angiogenesis. Other mechanisms by which ER stress can inhibit angiogenesis or cause aberrant angiogenesis include secondary OS, apoptosis, sterile inflammation, and cellular senescence. As adequate angiogenesis is critical to alveolar formation, we can expect ER stress to contribute to BPD.

### 5.2. ER Stress and Autophagy

Both ER stress and autophagy are stress response pathways that maintain cellular homeostasis. Under stress, the two pathways crosstalk and mutually activate each other to maintain an orchestrated response [138]. They also work together to limit the pro-inflammatory response [139]. If, however, the stress is excessive or sustained, these two pathways will jointly kill the cells by stabilizing the apoptosis effector PERP [140].

### 5.3. ER Stress and Cellular Senescence

Cellular senescence is a biological phenomenon first described by Hayflick and Moorhead six decades ago [141]. The topic recently attracted much attention due to its relationship with the aging process [142]. Cellular senescence is not the same as aging; it also appears during fetal development and controls organ patterning [143]. Senescence is also required during wound healing [144]. A higher level of senescence is detected in the lungs during the late saccular stage [145,146]. The timely removal of senescent cells by phagocytic cells is the key for the orchestrated organ development and wound healing. Any stress that leads to DNA damage can initiate cellular senescence. OS, a significant contributor to BPD, is one of the most common mechanisms to elicit senescent change. Strong evidence suggests that ER stress tightly interacts with cellular senescence, but their temporal relationship remains to be determined [147].

Behaviors of senescent cells include arrest proliferation, metabolic dysfunction, releasing inflammatory mediators, hypermetabolism, and transforming neighboring cells into senescent cells through the paracrine effect [148]. In the HOX rat BPD model, we showed a senescent change in multiple lung cell types, including AT2 and endothelial cells [21] (Figure 6). These findings undoubtedly indicate that cellular senescence contributes to BPD. TUDC and KYC treatments can decrease ER stress [16], attenuate cellular senescence [21], and improve alveolar formation in the HOX BPD rat lungs. These encouraging findings suggest that ER stress can be a therapeutic target for BPD treatment.

### 5.4. ER Stress in BPD Destructive Cycle

Based on our recent findings initiated by OS, we have built a description of the BPD destructive cycle (Figure 7). OS induces sterile inflammation with myeloid cell infiltration and MPO upregulation in the neonatal lungs, which results in a self-perpetuated destruction cycle involving ER stress and cellular senescence [9]. The ER stress fuels the destructive cycle by generating more OS from refolding proteins and electron transport chain uncoupling from mitochondrial dysfunction. The inflammation and nutrient-deprivation activity of senescent cells will only aggravate the cycle more. We hypothesize that this is the reason why therapy targeting only one BPD mechanism may not be enough to reduce the BPD prevalence.

### 5.5. Opportunity to Attenuate ER Stress for Treating BPD

Unlike cellular senescence, which does not increase at the lung saccular stage but persists after BPD rat pups have returned to room air [21], ER stress occurs as early as the lung saccular stage by OS [16]. The temporal relationship between ER stress and cellular senescence suggests that ER stress contributes mainly to BPD onset, while cellular senescence contributes to BPD progression. That is to say that the state of the disease should determine the BPD treatment. Chemical chaperones might be efficacious in the early BPD stage, when patients are still under oxygen treatment, while senotherapy might be appropriate during the later stage of BPD. Compared to chemical chaperones and senotherapeutics, KYC, as a systems pharmacology agent, provides therapeutic efficacy, expanding from the onset to the progression of BPD, which may be a better therapeutic agent for BPD treatment.

### 5.6. Potential Therapies for ER Stress

Chemical chaperones are the most commonly used compounds for treating ER stress. Still, some compounds attenuate ER stress without stabilizing unfolded proteins directly. 4-PB, TUDC, and guanabenz are the only three FDA-approved chemical chaperones, and pioglitazone is the only FDA-approved non-chemical chaperone ER stress suppressor that can be used in humans. However, they have not been approved for treating human proteostasis perturbation.

Animal and human evidence indicates ER stress contributes to a wide variety of diseases, including diabetes [149], neurodegenerative disease (Alzheimer’s disease, Parkinson’s disease, and Huntington’s disease) [150], asthma [151], pulmonary fibrosis [80], cystic fibrosis [152], α-1 antitrypsin deficiency [153], chronic obstructive pulmonary disease [154], et al. Reports show that chemical chaperones or pioglitazone treatments might benefit proteostasis perturbation diseases.

TUDC is efficacious in treating diabetes [155], neurodegenerative disorders [150], and pulmonary fibrosis [156]. In mice, TUDC has been shown to prevent progression from prediabetes to diabetes [157,158], diabetic visual deficits [159], and endothelial dysfunction [160,161]. TUDC has also been shown to attenuate amyloid deposition in mouse models of Alzheimer’s disease [162]. 4-PB is also known to have benefits for diabetes [163]. There are extensive investigations into using chemical chaperones to treat proteostasis perturbations caused by destabilizing missense mutations [95]. We should remember that some compounds do not have the properties of chemical chaperones but can attenuate ER stress by different mechanisms [110]. These compounds include crocin, proanthocyanidins and anthocyanin, cordycepin, pioglitazone, and caffeic acid phenethyl ester.

A search on ClincalTrials.gov found 333 registered trials of chemical chaperones or ER stress suppressors on diabetes (322 for pioglitazone, 5 for TUDC, 4 for 4-PB, and 1 for crocin), 7 registered trials for cystic fibrosis (3 for pioglitazone, 3 for 4-PB, and 1 for TUDC), 6 registered trials for asthma (5 for pioglitazone and 1 for TUDC), 5 registered trials for Alzheimer’s disease (3 for pioglitazone, 1 for TUDC, and 1 for 4-PB), 3 registered trials for pulmonary fibrosis (all for pioglitazone), and 1 registered trial for Parkinson’s disease (pioglitazone), but no clinical trial for α-1 antitrypsin deficiency, COPD, and BPD. We are not surprised that there are no registered trials in ClincalTrials.gov or reported clinical studies for BPD using chemical chaperones or ER stress suppressors, as the role of ER stress in BPD has only been suspected just recently.

## 6. Discussion

The extensive interaction with almost all subcellular organelles has made the ER critical in cellular homeostasis. Its multitasking ability also makes it highly susceptible to endogenous and exogenous stress. The most commonly described stress that disrupts ER function is OS. OS elicits ER stress (or UPR), the focus of this review, with a myriad of downstream responses. Excessive ER stress has been described to play a role in multiple lung disorders, including HOX-induced lung injury [20]. Using rodent models, our group and a few others have demonstrated that ER stress plays a role in developing the HOX-induced BPD phenotype. We showed that early postnatal caffeine treatment attenuates ER stress in the HOX rat BPD lungs and alveolar simplification [116]. Since conflicting effects of caffeine on ER stress have been reported in the literature, we started to ask the question about what the role of ER stress in BPD is. This review is based on a literature review and our research results. We hope the work can provide adequate information to interested researchers in related fields.

This review starts by describing why ER stress is considered in the pathophysiology of BPD. It is followed by a focused perspective of the structure of the ER and how several groups of specialized membrane-bound proteins maintain its complex structure. Those specialized proteins mainly stabilize the high curvature tubular form of the ER. The first section of the second part of this review describes the primary biological functions of the ER, including protein synthesis, lipid and cholesterol synthesis, calcium regulation, detoxification, and several inter-organelle communications. The third part briefly describes a particular type of autophagy, ER-phagy, that pertains to the ER.

The third part of this review starts by summarizing ER stress and then describes pathways after it. The endogenous chaperone BiP/GRP78 is the most crucial modulator of ER stress. Under non-stressed conditions, BiP/GRP78 chaperones the ER stress sensors (IRE1α, PERK, and ATF6) to prevent dimerization, phosphorylation, and cleavage. In stressed conditions, the ER cannot handle protein folding, and this proteostasis perturbation forces BiP/GRP78 to leave the sensors to assist protein refolding. As protein refolding requires an oxidative environment, ER stress will generate OS that amplifies the stress. IRE1α and PERK start homodimerization and phosphorylation after freeing from BiP/GRP78 to activate the downstream cascades to halt the synthesis of non-essential proteins and speed up the translation of BiP/GRP78 to resume ER homeostasis. The liberated ATF6 will be cleaved by the Golgi apparatus into a transcription factor (Figure 1) that prevents the protein synthesizing machinery from producing non-essential proteins. When proteostasis perturbation becomes unmanageable, excessive ER stress will initiate damaging processes, including inflammation, autophagy, apoptosis, ERAD, RIDD, and generating more OS.

The fourth part of the review is about chemical chaperones. Chemical chaperones are small molecules that can stabilize proteins non-specifically in specific harsh environments. Although some chemical chaperones have been studied in proteostasis disorders, no attempt has been made in the pediatric field. 4-PB, UDC, and TUDC are three chemical chaperones studied in children for diseases other than proteostasis disorders. Animal studies showed encouraging results in attenuating ER stress and BPD (Figure 4). We believe chemical chaperones may have potential in BPD treatment.

The last part of this review starts by addressing evidence suggesting ER stress contributes to BPD. This evidence includes how ER stress impairs angiogenesis, eliciting autophagy and cellular senescence, which culminates in a BPD destructive cycle we proposed recently (Figure 7) [9]. The complexity of the stakeholder relationship explains why a single-agent therapeutic strategy does not work for BPD. The temporal relationship between ER stress and cellular senescence suggests their contribution to BPD differs at different disease stages. Our novel reversible MPO inhibitor—KYC—can attenuate ER stress and cellular senescence in BPD lungs. Besides the antioxidant and anti-antiapoptotic activities, KYC also preserves the AT2 cell count in BPD lungs. Our data strongly support KYC as a systems pharmacology agent [128] with great potential in BPD treatment.

The considerations necessary for determining whether specifically targeting ER stress in BPD is a viable approach are numerous.
(i)If ER stress is a downstream event, targeting it may not resolve the underlying processes driving BPD, making it less valuable as a therapeutic target.(ii)Causal vs. Consequential in Complex Pathways: Causal and consequential factors are not mutually exclusive in many diseases. Just because a process like ER stress arises as a consequence of other pathological events does not mean that addressing it would be ineffective. For example, even if inflammation and OS are the primary drivers of BPD, ER stress may still amplify the damage or serve as a critical mediator of cellular dysfunction. Therefore, therapies targeting ER stress might mitigate disease progression even if they do not fully address the initial cause.(iii)Therapeutic Efficacy in Reducing Consequences: Even if ER stress is a consequence of BPD, it could still play a significant role in worsening disease severity by contributing to cell death, tissue damage, failure of repair processes, or even cellular senescence. In other diseases, such as neurodegenerative disorders, addressing downstream consequences (like protein misfolding or OS damage) has shown therapeutic value. Therefore, such an analysis is essential, since targeting a consequential process might still provide a therapeutic benefit by dampening a destructive feedback loop.(iv)Multifactorial Disease Processes: BPD likely involves multiple interacting processes, like many complex diseases. Even if ER stress is not a primary driver, it may synergize with other pathological mechanisms, such as mitochondrial dysfunction or immune activation. Therapeutic approaches targeting multiple points in the disease cascade, including causes and consequences, often provide better outcomes than focusing on a single factor.(v)The Importance of Timing: If ER stress is a consequence, the timing of intervention becomes essential. Early intervention targeting upstream processes might prevent ER stress altogether, whereas targeting ER stress later in the disease course could still alleviate symptoms or slow progression. Any analysis should factor in the stage of disease progression when evaluating whether targeting ER stress remains viable.

## 7. Conclusions

The frustratingly persistent prevalence of BPD has pressed us to seek new therapeutic strategies. Combining our literature search and our studies, we firmly believe ER stress contributes significantly to BPD. So many signaling pathways are involved in BPD that treatment focusing on one pathway will not be enough. The root causes of BPD are OS and inflammation [97]. Unfortunately, premature neonates susceptible to BPD are born with increased OS and inflammation. Supplemental oxygen treatment and mechanical ventilatory support are life-saving for premature neonates. The ER stress results from OS and inflammation, which reciprocally augments OS and inflammation; we thus consider ER stress a reasonable therapeutic target for premature neonates. Although our data suggest that ER stress has an impact probably in the onset stage of BPD [16], the employment of specific chemical chaperones should benefit neonatal lungs when premature neonates are still under oxygen support. The interactions between ER stress and other contributors suggest that chemical chaperones may still provide protection. Senotherapeutics may be helpful during BPD progression, as cellular senescence is not detectable at the saccular stage [21]. KYC, which holds tremendous potential as a systems pharmacology agent, offers more protection than chemical chaperones or senotherapeutics. This novel tripeptide repurposes MPO into a catalase-like protein and upregulates NRF2-mediated antioxidative enzymes [22] that attenuate ER stress [16]. We coined the term ‘systems pharmacological agent’ for KYC due to its multi-layered protection against BPD. Transcriptomic studies show that KYC reverses key changes in hyperoxic BPD rat lungs, including increased leukocyte migration, chemotaxis, degranulation, and NETosis, while restoring WNT-catenin and Notch signaling (unpublished data). Although KYC seems promising in our animal studies, it has yet to be FDA-approved. Due to the size of the rat pups, the route of administration was intraperitoneal. Other routes of administration need to be investigated. Presently, KYC is under NIH-funded phase II SBIR animal studies for toxicology and routes of administration. We hope other systems pharmacology agents will soon be available for BPD treatment.

## Figures and Tables

**Figure 1 cells-13-01774-f001:**
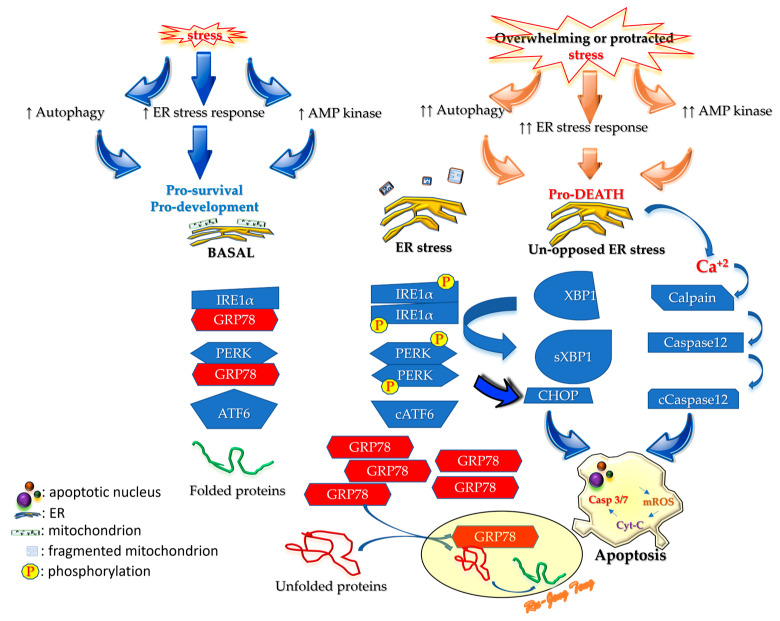
Endoplasmic reticulum (ER) stress or unfolded protein response (UPR). Stress distorting correct protein folding in the ER can elicit the ER stress response. Basal ER stress will upregulate the synthesis of the endogenous chaperone (BiP/GRP78) that assists protein refolding. Basal ER stress also inhibits the synthesis of non-essential proteins or degrades them so that raw material can be generated to synthesize essential proteins. The basal ER stress response is a survival mechanism for cells (left panel). If, however, the stress is overwhelming, BiP/GRP78 will all leave the ER stress sensors (IRE1α, PERK, and ATF6) to cope with the unfolded proteins. The protein refolding will generate reactive oxygen species that aggravate oxidative stress (OS) and lead to cell death. AMP: adenosine monophosphate; BiP: binding immunoglobulin protein; GRP78: glucose-regulated protein 78; IRE1α: inositol-requiring enzyme 1α; PERK: protein kinase R-like ER kinase; ATF6: activating transcription factor 6; cATF6: cleaved ATF6; mROS: reactive oxygen species from mitochondria; Cyt-C: cytochrome C; Casp 3/7: caspase 3 and 7; CHOP: C/EBP homologous protein; XBP1: X-box binding protein 1 (XBP1); sXBP1: split XBP1.

**Figure 2 cells-13-01774-f002:**
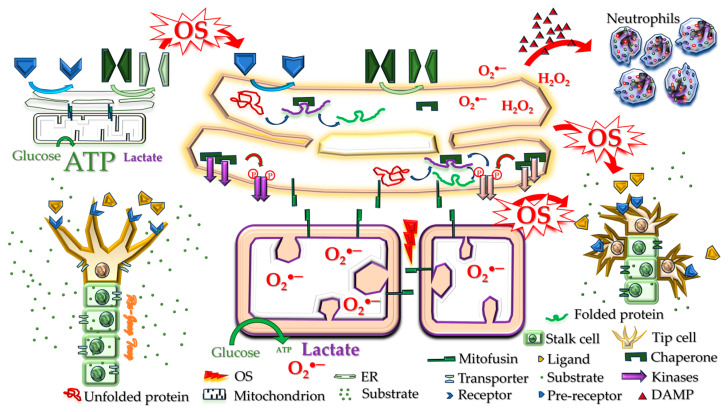
ER stress-associated responses that negatively affect angiogenesis. The ER intimately assists mitochondrial function in normal conditions and adequately processes protein post-translation modifications. Adequate oxidative phosphorylation and growth factor receptor function can produce appropriate angiogenesis critical for alveolar formation in neonatal lungs (left panel). Under excessive ER stress, growth factor receptors and substrate transporters cannot be adequately modified. The increased protein refolding generates many reactive oxygen species (ROS). Mitochondria distance themselves from the ER, so the electron transport chain becomes uncoupled, generating ROS. The accumulated OS then results in cell death that releases DAMPs to recruit neutrophils. The sterile inflammation, impaired oxidative phosphorylation, and dysfunctional growth factor receptors culminate in dysangiogenesis and impaired alveolar formation in neonatal lungs. OS: oxidative stress; DAMPs: damage-associated molecular patterns.

**Figure 3 cells-13-01774-f003:**
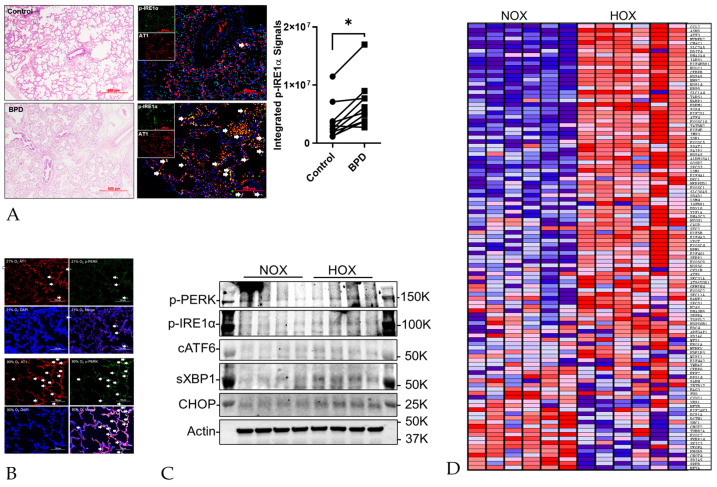
ER stress is detected in BPD lungs. (**A**) Immunofluorescence stain of phospho-IRE1α (green) and alveolar type 1 cell (AT1, red) shows increased colocalization in human BPD lungs. (**B**) Immunofluorescence stain of phospho-PERK (green) and AT1 (red) shows increased colocalization in hyperoxia (HOX)-exposed rat pup lungs. (**C**) Western blots show ER stress markers are upregulated as early as postnatal day 4 (P4). (**D**) Heatmap of the mRNA expression shows ER stress-related mRNAs increase in HOX-exposed rat pup lungs at P10. (The figure is modified from [16] under the Creative Commons CC BY 4.0 license). *: *p* < 0.05.

**Figure 4 cells-13-01774-f004:**
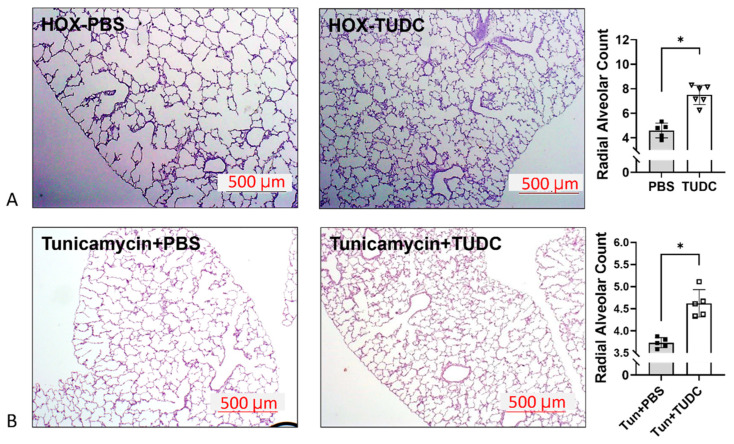
Taurodeoxycholate (TUDC) attenuates the alveolar simplification by HOX and tunicamycin. (**A**) The decreased radial alveolar counts in HOX rat lungs are increased in the group treated with 100 mg/kg/day TUDC. (**B**) The decreased radial alveolar counts in 0.1 mg/kg tunicamycin-treated rat lungs are increased in the group treated with TUDC. (The figure is reproduced from [16] under the Creative Commons CC BY 4.0 license.) Bar = 500 µm; *: *p* < 0.05, n = 5–6.

**Figure 5 cells-13-01774-f005:**
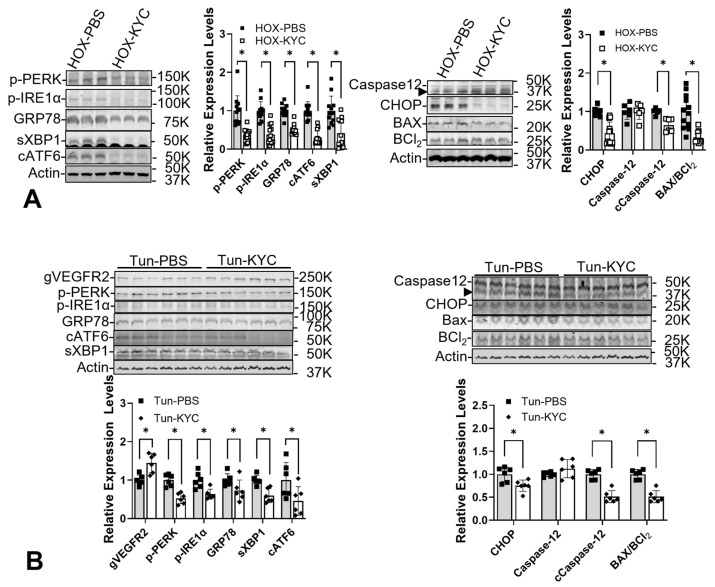
*N*-acetyl-lysyltyrosylcysteine amide (KYC) reduces ER stress caused by HOX and tunicamycin injection. (**A**) All the markers for ER stress in HOX-induced BPD in rat lungs are decreased by a daily KYC injection. (**B**) All the markers for ER stress in 0.1 mg/kg tunicamycin (TUN)-induced BPD in rat lungs are decreased by a daily injection of 10 mg/kg KYC. (The figure is reproduced from [16] under the Creative Commons CC BY 4.0 license.) *: *p* < 0.05, n = 6–12.

**Figure 6 cells-13-01774-f006:**
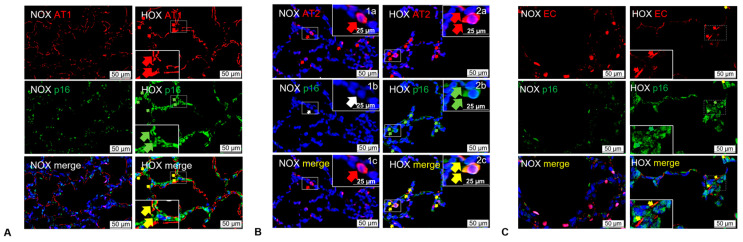
Cellular senescence is detected in multiple lung cell types of BPD rat lungs. The p16 (INK4/CDKN2A) antibody detects cellular senescence, the mouse anti-rat RT140 antibody detects AT1 cells, and the mouse anti-rat-endothelial-cell-antigen-1 (RECA-1) antibody detects rat endothelial cells. (**A**) Immunofluorescence stain shows colocalization of rat BPD lungs’ AT1 stain (red) and p16 stain (green). (**B**) Colocalization is seen between the AT2 stain (surfactant protein B, red) and p16 (green). (**C**) Colocalization is seen between endothelial cells and p16. (The figure is reproduced from [21] under permission obtained from the American Thoracic Society.) The arrows indicate specific cells identified.

**Figure 7 cells-13-01774-f007:**
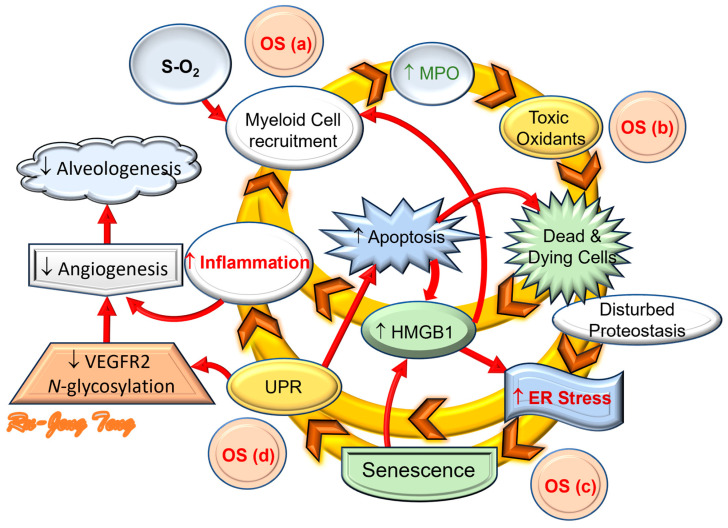
The intertwined interaction between inflammation and OS forms the destructive cycle of BPD. Oxidative stress (OS) from oxygen treatment is the initiating event (OS(a)) in BPD, which recruits myeloperoxidase (MPO)-containing myeloid cells (macrophages and neutrophils) to infiltrate the alveolar sacs. MPO released from the myeloid cells then generates hypochlorous acid (HOCl), a potent reactive oxygen species (ROS), as the second source of OS (OS(b)). The unopposed OS perturbates the proteostasis to elicit endoplasmic reticulum (ER) stress with a subsequent third source of OS (OS(c)). The ER stress contributes to cellular senescence or unfolded protein response (UPR) with another wave of OS (OS(d)). The high-mobility group box 1 (HMGB1) from the senescence-associated secretory pattern (SASP) and UPR induce sterile inflammation to perpetuate the inflammatory process and OS. ER stress will inhibit the glycosylation of vascular-endothelial-cell-growth-factor receptor 2 (VEGFR_2_) and, with the sterile inflammation, will inhibit angiogenesis in neonatal lungs with poor alveolar formation. (The figure is modified from [9] under the Creative Commons CC BY 4.0 license.) ↑: increase; ↓: decrease.

**Table 1 cells-13-01774-t001:** ER stress in other disease conditions. Disorders are grouped according to the involved subcellular organelle under the same highlight color. Some disorders (diabetes in red, Parkinson’s disease in blue, and intrahepatic cholestasis in green) may involve multiple organelles, and the same font color is assigned to each. (Table reproduced from [95] under the Creative Commons CC BY 4.0 license).

Disease	Gene	Uniprot Code	Protein Type	Subcellular Location	# of Articles
Gaucher	*GBA*	P04062	Enzyme	Lysosome	64
Fabry	*GLA*	P06280	Enzyme	Lysosome	42
GM-1, Morqio B	*GLB1*	P16278	Enzyme	Lysosome	16
Pompe	*GAA*	P10253	Enzyme	Lysosome	14
Cystic fibrosis	*CFTR*	P13569	Transporter	Plasma membrane	14
Retinitis pigmentosa	*RHO*	P08100	Receptor	Plasma membrane	12
Phenylketonuria	*PAH*	P00439	Enzyme	Cytosol	9
Krabbe disease	*GALC*	P54803	Enzyme	Lysosome	9
Nephrogenic diabetes insipidus	*V2R*	P30518	Receptor	Plasma membrane	8
Long QT syndrome	*KCNH2*	Q12809	Transporter	Plasma membrane	7
**Parkinson’s**	*PARK7*	Q99497	Enzyme	Plasma membrane, nucleus, mitochondrion	5
Niemann–Pick	*NPC1*	O15118	Receptor	Lysosome	5
Hyperoxaluria	*AGXT*	Q86XE5	Enzyme	Mitochondrion	5
Obesity	*MC4R*	P32245	Receptor	Plasma membrane	4
GM-2, Sanfilippo syndrome	*GNRHR*	P07686	Enzyme	Lysosome	4
GM-2, Tay–Sachs syndrome	*HEXB*	P06865	Enzyme	Lysosome	4
Galactosemia	*HEXA*	P07902	Enzyme	Cytosol	4
Hypoparathyroidism	*PTH*	P01270	Hormone	Extracellular or secreted	3
**Parkinson’s**	*GALT*	P04062	Enzyme	Lysosome	3
Hypogonadotropic hypogonadism	*ATP7B*	P30968	Receptor	Plasma membrane	2
Wilson’s disease	*PMM2*	P35670	Transporter	Golgi apparatus	2
PMM2-CDG	*SLC26A4*	O15305	Enzyme	Cytosol	2
Pendred	*MMAB*	O43511	Transporter	Plasma membrane	2
Methylmalonic aciduria	*ABCB4*	Q96EY8	Enzyme	Mitochondrion	2
**Intrahepatic cholestasis**	*DRD4*	P21439	Transporter	Plasma membrane	2
Hyperactivity disorder	*ABCC8*	P21917	Receptor	Plasma membrane	2
**Diabetes**	*GPR56*	Q09428	Receptor	Plasma membrane	2
Polymicrogyria	*PGK1*	Q9Y653	Receptor	Plasma membrane, extracellular, or secreted	1
Phosphoglycerate kinase 1 deficiency	*SNCA*	P00558	Enzyme	Cytosol	1
**Parkinson’s**	*SUMF1*	P37840	Regulator	Presynaptic vesicle	1
Multiple sulfatase deficiency	*NPM*	Q8NBK3	Enzyme	ER	1
Leukemia	*PKR2*	P06748	Regulator	Nucleus, cytoskeleton	1
**Intrahepatic cholestasis**	*ABCB11*	O95342	Transporter	Plasma membrane	1
Nocturnal frontal lobe epilepsy	*CHRNB2*/*CHRNA4*	P17787/P43681	Transporter	Plasma membrane	1
Hypomagnesemia	*CLDN16*	Q9Y5I7	Transporter	Plasma membrane	1
Creutzfeldt–Jakob, Kuru	*PRNP*	P04156	Unclear/prion	Plasma membrane	1
Homocystinuria	*CBS*	P35520	Enzyme	Nucleus	1
Fibrodysplasia ossificans	*ACVR1*	Q04771	Enzyme	Plasma membrane	1
Epilepsy, migraine	*SCN1A*	P35498	Transporter	Plasma membrane	1
Dystonia	*SLC2A1*	P11166	Transporter	Plasma membrane	1
Diarrhea (cholera toxin)	*NHE3*	P48764	Transporter	Plasma membrane	1
**Diabetes**	*KCNJ11*	Q14654	Transporter	Plasma membrane	1
**Intrahepatic cholestasis**	*ATP8B1*	O43520	Transporter	Plasma membrane, Golgi apparatus, or ER	1
Breast cancer	*NBS*	O60934	Regulator	Nucleus	1
Amyotrophic lateral sclerosis	*SOD1*	P00441	Enzyme	Nucleus, mitochondrion	1

## Data Availability

The original contributions presented in the study are included in this article; further inquiries can be directed to the corresponding author.

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
