# Peer review of "Endoplasmic Reticulum Stress in Bronchopulmonary Dysplasia: Contributor or Consequence?"

_cells, 2024, doi:10.3390/cells13211774_

Round 1
Reviewer 1 Report
Comments and Suggestions for Authors
Bronchopulmonary dysplasia (BPD) is the most common pulmonary complication in premature infants, however its pathobiological processes remain poorly understood. Wu et al. reported the role of endoplasmic reticulum (ER) stress in BPD. This review describes the role of ER stress in BPD and discusses the therapeutic potentials of chemical chaperones and N-Acetyl-lysyltyrosylcysteine amide (KYC), a myeloperoxidase inhibitor that attenuates ER stress and senescence.
This is a very extensive review of ER, ER stress and its consequences. Potentials treatments that attenuate ER stress in general disease conditions and in BPD are well discussed.
Author Response
Comments 1: [Bronchopulmonary dysplasia (BPD) is the most common pulmonary complication in premature infants, however its pathobiological processes remain poorly understood. Wu et al. reported the role of endoplasmic reticulum (ER) stress in BPD. This review describes the role of ER stress in BPD and discusses the therapeutic potentials of chemical chaperones and N-Acetyl-lysyltyrosylcysteine amide (KYC), a myeloperoxidase inhibitor that attenuates ER stress and senescence. This is a very extensive review of ER, ER stress and its consequences. Potentials treatments that attenuate ER stress in general disease conditions and in BPD are well discussed.]
Response: Thank you so much for the positive comments.
Reviewer 2 Report
Comments and Suggestions for Authors
Wu and other authors draft this review to provide an overview of role that Endoplasmic Reticulum Stress (ER stress) play in bronchopulmonary dysplasia (BPD). The paper discusses the therapeutic potentials of chemical chaperones and KYC and highlighting their promising role in future therapeutic interventions. Since the relevant mechanisms of KYC treated BPD are investigated recently, even though the topic has already been discussed in several other reviews published this year by the same group (PMID: 39199135 https://doi.org/10.3390/antiox13080889, and PMID: 39337630 https://pmc.ncbi.nlm.nih.gov/articles/PMC11431892/ ), this review still shows a specific angle try to provide evidence that targeting ER stress could be a potential solution of BPD. This review is accomplished on summarizing ER, ER stress, Treatments that Attenuate ER Stress and how it works in BPD, which organized well and clear that include entire information, especially discusses deeply on why ER stress is considered in the pathophysiology of BPD. However, as the paper mentioned “If ER stress is a downstream event, targeting it may not resolve the underlying processes driving BPD, making it less valuable as a therapeutic target.” Reviewer what to ask why authors still choose to bring ER stress on the table since OS and inflammation shows stronger relevant to BPD? It seems like KYC treatment can cause a series of downstream change which may not be a good drug to study underline mechanism. I agree that it is interesting to think about it and may come up with some different treatments that may help our patient during the disease. That’s why I recommend published. But the question is we need a strong SIGNIFICANCE that authors may want to explain more about it. Overall, this is an excellent work.
Author Response
Comments 1: [Wu and other authors draft this review to provide an overview of role that Endoplasmic Reticulum Stress (ER stress) play in bronchopulmonary dysplasia (BPD). The paper discusses the therapeutic potentials of chemical chaperones and KYC and highlighting their promising role in future therapeutic interventions. Since the relevant mechanisms of KYC treated BPD are investigated recently, even though the topic has already been discussed in several other reviews published this year by the same group (PMID: 39199135 https://doi.org/10.3390/antiox13080889, and PMID: 39337630 https://pmc.ncbi.nlm.nih.gov/articles/PMC11431892/ ), this review still shows a specific angle try to provide evidence that targeting ER stress could be a potential solution of BPD. This review is accomplished on summarizing ER, ER stress, Treatments that Attenuate ER Stress and how it works in BPD, which organized well and clear that include entire information, especially discusses deeply on why ER stress is considered in the pathophysiology of BPD.]
Response 1: We appreciate your kind comments.
Comments 2: [However, as the paper mentioned “If ER stress is a downstream event, targeting it may not resolve the underlying processes driving BPD, making it less valuable as a therapeutic target.” Reviewer what to ask why authors still choose to bring ER stress on the table since OS and inflammation shows stronger relevant to BPD?]
Response 2: Thanks for pointing out the critical issue of why we brought out ER stress in this review. As we discussed in two prior relevant reviews, OS and inflammation are the root causes of BPD, and it seems modulating ER stress may not be effective. However, as neonatologists, we understand that oxygen therapy is unavoidable for most premature neonates. Antioxidants and anti-inflammatory therapies have not shown any efficacy. Because ER stress can augment OS and inflammation, we believe modulating ER stress has therapeutic potential. We HAVE ADDED SOME FURTHER DISCUSSION ON THIS MATTER TO CLARIFY OUR THINKING AND PERSPECTIVE. (lines 864-870).
Comments 3: [It seems like KYC treatment can cause a series of downstream change which may not be a good drug to study underline mechanism. I agree that it is interesting to think about it and may come up with some different treatments that may help our patient during the disease. That’s why I recommend published. But the question is we need a strong SIGNIFICANCE that authors may want to explain more about it. Overall, this is an excellent work.]
Response 3: We thank the reviewer for bringing this point to our attention. We did not clarify that our understanding of the mechanisms of action for KYC allows us to use it as a multimodal probe for complex disease conditions such as BPD. We have highlighted this approach in our research papers (Jing X, et al. Cellular Senescence Contributes to the Progression of Hyperoxic Bronchopulmonary Dysplasia. Am J Respir Cell Mol Biol. 2024 Feb;70(2):94-109. and Teng RJ, et al. N-acetyl-lysyltyrosylcysteine amide, a novel systems pharmacology agent, reduces bronchopulmonary dysplasia in hyperoxic neonatal rat pups. Free Radic Biol Med. 2021 Apr;166:73-89. ). Both papers are cited in this review as Ref 21 and Ref 22. We have added text in the amended review to address this matter (lines 877-883).